# Analysis of miRNA Expression in Patients with Rheumatoid Arthritis during Olokizumab Treatment

**DOI:** 10.3390/jpm10040205

**Published:** 2020-10-31

**Authors:** Irina V. Bure, Dmitry S. Mikhaylenko, Ekaterina B. Kuznetsova, Ekaterina A. Alekseeva, Kristina I. Bondareva, Alexey I. Kalinkin, Alexander N. Lukashev, Vadim V. Tarasov, Andrey A. Zamyatnin, Marina V. Nemtsova

**Affiliations:** 1Institute of Molecular Medicine, Sechenov First Moscow State Medical University (Sechenov University), Trubetskaya str., 8-2, 119992 Moscow, Russia; bureira@mail.ru (I.V.B.); dimserg@mail.ru (D.S.M.); kuznetsova.k@bk.ru (E.B.K.); ekater.alekseeva@gmail.com (E.A.A.); alexander_lukashev@hotmail.com (A.N.L.); 2Laboratory of Epigenetics, Research Centre for Medical Genetics, Moskvorechye str. 1, 115478 Moscow, Russia; alexeika2@yandex.ru; 3Biostatistics Department, OCT Rus, Bolshaya Moskovskaya str., 8/2, 191002 Saint-Petersburg, Russia; kbondareva@oct-clinicaltrials.com; 4Martsinovsky Institute of Medical Parasitology, Tropical and Vector Borne Diseases, Sechenov First Moscow State Medical University, 119435 Moscow, Russia; 5Department of Pharmacology and Pharmacy, Sechenov First Moscow State Medical University, 119991 Moscow, Russia; tarasov-v-v@mail.ru; 6Belozersky Institute of Physico-Chemical Biology, Lomonosov Moscow State University, 119992 Moscow, Russia; 7Department of Biotechnology, Sirius University of Science and Technology, 1 Olympic Ave, 354340 Sochi, Russia

**Keywords:** rheumatoid arthritis, microRNA, biomarker, personalized medicine

## Abstract

Rheumatoid arthritis (RA) is the most common autoimmune disease worldwide. Epigenetic alternations of microRNAs (miRNAs) can contribute to its pathogenesis and progression. As the first line therapy with DMARDs is not always successful, other drugs and therapeutic targets should be applied. This study aims to measure the expression level of plasma miRNAs in RA patients treated with olokizumab and to evaluate their potential as prognostic biomarkers. The expression of 9 miRNAs was quantified in 103 RA patients before treatment and at weeks 12 and 24 of olokizumab therapy by reverse transcription-polymerase chain reaction (RT-PCR) assay and analyzed in groups of responders and non-responders. Almost all miRNAs changed their expression during therapy. The ROC curve analysis of the most prominent of them together with consequent univariate and multivariate regression analysis revealed statistically significant associations with the olokizumab therapy efficiency scores for miR-26b, miR-29, miR-451, and miR-522. Therefore, these miRNAs might be a potential therapeutic response biomarker.

## 1. Introduction

Rheumatoid arthritis (RA) is a chronic autoimmune disease associated with systemic inflammation that leads to severe complications, joint destruction, and reduced mobility. RA affects 1% of the world’s population, with a higher prevalence in Europeans and Asians [1]. RA is characterized by a significant clinical polymorphism with a wide variability of symptoms, clinical forms, and prognoses [2].

The first-line therapy for moderate to severe RA is conventional disease-modifying antirheumatic drugs (DMARDs) with methotrexate (MTX) being the most commonly used [3]. Biologic DMARDs are targeting inflammatory cytokines such as tumor necrosis factor (TNF), leading to normalization of their levels, and the alleviation of clinical symptoms. However, around 45–50% of patients demonstrate resistance to this treatment and have to switch therapy in the first year [4]. Thus, alternative RA therapeutics are still needed for the large population of patients who have experienced treatment failure of conventional or biologic DMARDs.

The relatively new group of drugs, which includes agents targeting the IL-6, demonstrate promising results. IL-6 is an important mediator of inflammation that is involved in immunologic processes underlying RA pathogenesis, such as T-cell activation and B-cell proliferation. The amount of IL-6 in the synovial fluid of patients with RA has been shown to correlate with the severity of synovitis and joint destruction [5].

Olokizumab (OKZ, CDP6038) is a humanized anti-IL-6 monoclonal antibody. It targets the IL-6 cytokine rather than the receptor, and selectively blocks the final assembly of the signaling complex. In Phase I (healthy volunteers), II, and III (patients with moderate and severe RA on MTX) clinical trials, OKZ was effective in alleviating RA symptoms, well tolerated after intravenous and subcutaneous delivery with a median plasma half-life of approximately 31 days, 63% bioavailability, and no apparent antidrug-antibody-mediated clearance [6].

Divergence in clinical forms of RA and suitable treatment, at least in part, could be explained by epigenetic factors, including microRNA (miRNA). MiRNAs are evolutionarily conserved small non-coding transcripts of 18–25 nt that regulate gene expression at the post-transcriptional level by either repressing the translation or causing degradation of multiple target mRNAs [7]. The role of miRNA in the inflammatory process includes both control of cytokine production and protection of cartilage tissue by regulating catabolic activity, proliferation, and resistance to apoptosis [8]. Currently more than 100 miRNAs have been identified that are potentially capable of affecting the molecular pathways that control development and functions of immune cells [9]. A number of them could be regarded as potential biomarkers, because of their aberrant expression in pathologic conditions, high tissue-specificity, and presence in biological fluids. Moreover, some miRNAs are differentially expressed in response to drug treatment and therefore can serve as predictive factors for the clinical response to therapies among RA patients [2].

Therefore, in our study, we explore nine candidate miRNAs involved in the pathogenesis of RA (miR-29, miR-16, miR-155, miR-20a, miR-451, miR-26b, miR-522, miR-192, and miR-137) in the plasma samples of RA patients, treated with olokizumab. Comparing the groups of responders and non-responders to olokizumab treatment, we aimed to assess the efficacy of miRNAs as promising non-invasive biomarkers to predict the response of RA patients to the therapy.

## 2. Materials and Methods

### 2.1. Patients

The study involved 103 RA patients, treated with olokizumab in the clinical study number CL04041022 “A randomized, double blind, parallel group, placebo controlled, multicenter phase III study of the efficacy and safety of olokizumab in subjects with moderately to severely active rheumatoid arthritis inadequately controlled by methotrexate therapy.”

The inclusion criteria were: patients with 18 years and older who have moderately to severely active RA disease and inadequate response to treatment with MTX for at least 12 weeks prior to screening at a dose of 15 to 25 mg/week (or ≥10 mg/week if intolerant to higher doses). The key exclusion criteria were: patients with any other inflammatory diseases or serious medical condition, such as infections; pregnant or lactating women; patients exposed to any licensed or investigational compound directly or indirectly targeting IL6 or IL6R; treatment with cell depleting therapies including anti CD20 or conventional DMARDs, other than MTX.

The current study was performed on samples from Russian patients. All subjects gave their informed consent for inclusion before they participated in the study. The study was conducted in accordance with the Declaration of Helsinki, and the protocol was approved by the Ethics Committee of Sechenov First Moscow State Medical University (Sechenov University), 13 July 2016.

### 2.2. Procedures

Olokizumab was injected subcutaneously at a dose of 64 mg every 4 weeks (q4w) or 64 mg every 2 weeks (q2w) during 24 weeks. The primary endpoint was response rate at week 12 based on 20% improvement in American College of Rheumatology criteria (ACR20), with further study of subjects over at least a 24-week period.

Blood samples from each patient were collected into K3EDTA tubes pre-dose, at week 12 of treatment with olokizumab and, when relevant, at week 24. Plasma was obtained by centrifugation.

### 2.3. RNA Extraction and Reverse Transcription-Polymerase Chain Reaction (RT-PCR) Assay

Total RNA, including miRNA, was extracted from plasma samples using Trizol (Life Technologies, Carlsbad, CA, USA) and a set of miRNeasy Mini Kit (Qiagen, Hilden, Germany) following the protocol of manufacturers with small modifications. Concentration and purity of the obtained RNA was estimated on the NanoDrop 2000 microvolume spectrophotometer (Thermo Fisher Scientific, New York, NY, USA). The extraction process was repeated for each sample until a sufficient amount of RNA was obtained for the next steps.

cDNA was synthesized from 300 ng of total RNA with MiScript II RT Kit (Qiagen) under the recommended protocol. Real-time PCR was performed on the CFX96 Real-Time PCR Detection System (Bio-Rad, Hercules, USA) in three repetitions for each of the analyzed miRNAs, as well as spiked-in control cel-miR-39-3p for data normalization, using the MiScript SYBR Green PCR Kit (Qiagen) according to manufacturer’s recommended protocol. The primer sequences for miRNAs are listed in the Table 1. Presynthesized miScript Primer Assay (Qiagen) primer was used for control. Data were analyzed using the ΔΔCt method.

### 2.4. Prediction of miRNA Targets

The associations between miRNAs and genes of signaling pathway IL-6/IL-6R were searched by using the integrative database of human miRNA target predictions mirDIP [10], which aggregates the data from all known miRNA databases. The likelihood level was specified as high. The visualization of obtained results was performed in the Navigator program [11].

### 2.5. Statistical Analysis

Statistical analysis of the results was performed using the Statistical Analysis Systems (SAS) 9.4 (SAS Institute Inc., Cary, NC, USA).

The normality of sample distribution was evaluated using the Shapiro-Wilk test and graphical methods. The correlation between relative microRNA expression before treatment and clinical indices of disease activity, as well as other characteristics of patients was evaluated using the Spearman rank correlation coefficient with 95% CI. For dichotomous variables of therapy response and comparison of miRNA expression between responders and non-responders the nonparametric Mann–Whitney U test was used.

To investigate the potential of miRNAs as a prognostic biomarker, the receiver operating characteristic (ROC) curve was generated, and the area under the curve (AUC) was calculated by computing sensitivity and specificity at various threshold levels. Potentially useful predictors were included into univariate and multivariate logistic regression analysis.

*P*-values < 0.05 were considered statistically significant.

## 3. Results

### 3.1. Clinical Features and General Characteristics of the Patients

Totally 103 RA patients (90 females and 13 males) with the mean age of 51.8 ± 12.17 years were recruited in the study. The median duration of the disease was 6.29 years, and the majority of patients (91.3%) had high disease activity (DAS28 > 5.1).

The initial mean value of Disease Activity Score 28-joint Count (DAS28) was 6.03 ± 0.676, Clinical Disease Activity Index (CDAI) was 40.47 ± 9.064, and Health Assessment Questionnaire-Disability Index (HAQ-DI) was 1.7339 ± 0.4882. The initial levels of C-reactive protein (CRP), anti-cyclic citrullinated peptide antibodies (anti-CCP) and rheumatoid factor (RF) were 23.5 ± 22.85 mg/mL, 531.35 ± 671.132, and 193.1 ± 224.15, respectively. The level of anti-CCP > 10 IU/mL was observed in 83.5% patients, the level of RF ≥ 15 IU/mL was observed in 87.4%. The patients’ demographics and clinical characteristics are summarized in the Table 2.

All the patients included in the analysis received either olokizumab q4w (55 patients, 53.4%) or q2w (48 patients, 46.6%), depending on the randomization schedule of the core clinical study. The efficacy of olokizumab therapy was assessed at weeks 12 and 24.

In patients of this population, the mean change in DAS28-CRP was −2.53 ± 1.083 by week 12 and −2.85 ± 1.196 by week 24. Based on the DAS28-CRP, the low disease activity (<3.2) was achieved in 42/103 (40.8%) patients by week 12, and in 52/103 (50.5%) patients by week 24. According to ACR20, the response was achieved in 71/103 (68.9%) patients by week 12 and in 83/103 (80.6%) by week 24, and response based on 50% improvement according to ACR criteria (ACR50) was achieved in 48/103 (46.6%) and 56/103 (54.4%), respectively.

Therefore, following the olokizumab therapy, the patients demonstrate significant differences of clinical and laboratory parameters, that enables to subdivide them into groups of responders and non-responders to olokizumab therapy based on the scores DAS28-CRP, ACR20, and ACR50 (Table 3).

### 3.2. MiRNA Expression Is Associated with Olokizumab TREATMENT

Nine miRNAs were chosen for the investigation according to the literature data, indicating their potential role in RA initiation and progression. To assess the changes in miRNAs expression, we quantified miRNAs abundance in plasma samples of RA patients before treatment with olokizumab and at the weeks 12 and 24 of therapy by using qRT-PCR (Table 4).

Analysis of their relative expression revealed statistically significant increase of miR-155, miR-16, miR-192, miR-20a, miR-29, and miR-451 level by week 12. However, their levels were decreased further, and there were no more significant differences compared to the baseline at week 24. The only exception was miR-451, which expression at week 24 was higher than at week 12, and the differences compared to the baseline were statistically significant. On the contrary, the expression of miR-26b and miR-522 was decreased at weeks 12 and 24.

### 3.3. MiRNA Expression Is Associated with Some Patient Characteristics

The miRNA abundance in patient samples before olokizumab treatment was also analyzed in relation to baseline characteristics of patients. Statistically significant correlations (*p* < 0.05) were determined between the baseline expression of miR-26b, miR-29 and the disease severity (*p* = 0.0486 and 0.0141, respectively), and between miR-522 and the presence of anti-CCP (*p* = 0.0447) (Table 5).

### 3.4. MiRNA Expression Is Associated with Olokizumab Therapy Efficiency

To evaluate potential correlations between miRNAs expression and olokizumab therapy efficiency, the baseline expression of each miRNA was compared in groups of responders and non-responders. MiRNAs demonstrated significant differences were subjected to further analysis and the receiver operating characteristic (ROC) curves were generated for each of them. Statistically significant results were obtained for the following associations: miR-29 and ACR20 response at week 12 and 24, ACR50 at week 12 and 24, and DAS28-CRP response at week 24; miR-451 and ACR20 response at week 12, ACR50 at week 12, DAS28-CRP response at week 12; miR-26b and ACR50 response at weeks 12 and 24, DAS28-CRP response at weeks 12 and 24; miR-522 and ACR50 response at week 12, DAS28-CRP response at week 12 (Figure 1; Appendix A).

Further, we performed univariate logistic regression analysis for miRNAs with area under the ROC curve (AUC) statistically significantly greater than 0.5. When miRNA level before olokizumab treatment was a significant risk factor at the 0.1 level in the univariate logistic regression analysis, it was included in the multivariate analysis, along with other significant factors at the 0.1 level, potentially associated with treatment outcomes (gender, age, body weight, disease duration, initial disease activity), using stepwise forward multivariate logistic regression analysis (Appendix A).

Univariate logistic regression did not reveal statistically significant associations between miRNA levels at baseline and ACR20 at week 12. However, the higher baseline miR-29 expression was correlated with lower odds of ACR20 response at week 24 (odds ratio (OR): 0.65, [95% confidence interval (CI)) 0.45–0.95], *p* = 0.0249).

Univariate logistic regression determined four miRNAs, whose high expression level in RA patients was associated with lower odds of ACR50 response at week 12, namely miR-26b (OR: 0.70 [95% CI 0.51–0.97], *p* = 0.0332), miR-29 (OR: 0.64 [95% CI 0.46–0.91], *p* = 0.0116), miR-451 (OR: 0.63 [95% CI 0.50–0.80], *p* = 0.0001), miR-522 (OR: 0.74 [95% CI 0.59–0.93], *p* = 0.0082). Multivariate logistic regression analysis confirmed significant association for miR-451 expression (OR: 0.63 [95% CI 0.50–0.80], *p* < 0.0001).

Univariate logistic regression revealed that higher expression of miR-26b (OR: 0.71 [95% CI 0.51–0.97], *p* = 0.0342) and miR-29 (OR: 0.65 [95% CI 0.47–0.91], *p* = 0.0131) were associated with a lower odds of ACR50 response at week 24. In the multivariate regression analysis, a significant correlation was demonstrated for miR-29 expression (OR: 0.65 [95% CI 0.47–0.91], *p* = 0.0131).

Univariate logistic regression revealed 3 miRNAs, whose higher expression level was associated with lower odds of DAS28-CRP response at week 12. These are miR-26b (OR: 0.70 [95% CI 0.50–0.98], *p* = 0.0365), miR-451 (OR: 0.76 [95% CI 0.62–0.92], *p* = 0.0059), and miR-522 (OR: 0.68 [95% CI 0.53–0.86], *p* = 0.0017). Multivariate logistic regression analysis confirmed significant association for miR-522 expression (OR: 0.68 [95% CI 0.53–0.86], *p* = 0.0017).

Univariate logistic regression revealed two miRNAs, whose higher expression level was associated with lower odds of DAS28-CRP response at week 24. These are miR-26b (OR: 0.51 [95% CI 0.35–0.75], *p* = 0.0006) and miR-29 (OR: 0.59 [95%CI 0.41–0.84], *p* = 0.0034). Multivariate logistic regression analysis confirmed significant association for miR-26b expression (OR: 0.50 [95% CI 0.33–0.74], *p* = 0.0005) and the factor “male gender” that was correlated with better clinical response (OR: 4.44 [95% CI 1.05–18.72], *p* = 0.0423).

Therefore, the ROC curve analysis together with the regression analysis revealed that miR-26b, miR-29, miR-451, and miR-522 might be a potential therapeutic response biomarker.

## 4. Discussion

Recently a number of studies revealed an important role of miRNAs in RA pathogenesis. Aberrant miRNA regulation occurs in various cells and tissues in RA [7]. The role of miRNA in the inflammatory process includes both control of cytokine production and protection of cartilage tissue by regulating catabolic activity, proliferation, and resistance to apoptosis. Taking into account that miRNAs are stable in plasma and synovial fluid and their differential expression correlates with RA stage and prognosis, miRNAs could be considered as potential biomarkers. Expression evaluation of different known miRNAs, circulating in plasma, can be helpful to identify the biomarkers with diagnostic capacity as well as predictors of the disease pathogenesis [12].

In the present study: we analyzed the literature data and chosen nine miRNAs miR-29, miR-16, miR-155, miR-20a, miR-451, miR-26b, miR-522, miR-192, and miR-137, characterized as functionally related to RA progression. Thus, miR-155 demonstrate pro-inflammatory activity [13] and regulates destructive processes in rheumatoid arthritis synovial fibroblasts (RASFs) because of the matrix metalloproteinases MMP-1 and MMP-3 expression repression [14], as well as miR-522 that regulate the expression of proinflammatory cytokines and MMPs via targeting SOCS3 [15]. Both miR-20a and miR-16 participate in regulation of apoptosis [16,17], whereas miR-451 and miR-192 regulate synovial fibroblasts proliferation [18,19]. The potential roles of all nine miRNAs in RA, according to the recent investigation are summarized in Table 6.

We found that almost all these miRNAs significantly change their abundance in plasma during treatment with olokizumab. However, most of them lack the effect already by the week 24 after start of therapy. Nevertheless, the baseline expression of miRNAs miR-26b, miR-29, miR-451, and miR-522 was significantly different, when compared in groups of responders and non-responders to the therapy. For each of them the higher expression indicated lower odds of clinical response to olokizumab treatment.

As miRNAs exert their functions by targeting mRNAs and negatively regulating translation, their effect depends on target’s functions [34]. The capability to target numerous effectors of important signaling pathways has made numerous miRNAs one of crucial genome regulators. As olokizumab acts as inhibitor of IL-6/IL-6R, we proposed that prognostic potential of these miRNAs in olokizumab treatment could be in part associated with their effect on the signaling pathway.

Interleukin-6 (IL-6) is a cytokine that provokes a broad range of cellular and physiological responses, including inflammation. To produce these effects IL-6 signals through a receptor composed of two subunits, a specific receptor for IL-6 and a cell-surface glycoprotein gp130 as a signal transducer. Binding of IL-6 to its receptor initiates activation of JAK kinases, which further phosphorylate and activate STAT transcription factors, especially STAT3, that move into the nucleus to activate transcription of genes containing STAT3 response elements. Elevated levels of IL-6 and IL-6R in the serum of RA patients are positively correlated with RA disease severity and radiological joint damage [4]. The prediction and analysis of miRNA targets was performed for miR-26b, miR-29, miR-451, and miR-522, confirmed our suggestions about existence of miRNA-gene target pairs in IL-6/IL-6R (Figure 2).

Previous investigations also confirm the role of this miRNAs in signaling pathway IL-6/IL-6R. For example, miR-451 targets IL-6R-STAT3-VEGF signaling and thus suppresses angiogenesis in hepatocellular carcinoma [35]. Moreover, all these miRNAs were previously mentioned as potential biomarkers, because of their aberrant expression in RA. Circulating miR-26b was described as significantly upregulated in patients with immune-mediated inflammatory disorders even on early stages, which enables to discriminate them from healthy controls [36,37] and as a prognostic biomarkers, associated with response to expanded allogenic adipose-derived mesenchymal stem cells (eASCs) in the RA patients [38]. Serum miR-29 was differentially expressed in some autoimmune disorders [39]. MiR-451 expression was elevated in the plasma and serum of RA patients, comparing with the healthy donors [40], however, was lower in synovial fibroblasts [18] and neutrophils [41]. miR-522 was upregulated in synovial fibroblasts from RA patients, and its expression was correlated with the RA-associated clinical parameters [15]. However, none of them, except of miR-26b, was suggested as prognostic factor and even in this case, it was not related to olokizumab treatment.

There are a number of drugs that inhibit IL-6 signaling pathway, participating in clinical trials or already applied in clinical practice. Monoclonal antibodies that bind IL-6 or its receptor demonstrate a high degree of specificity, target specific epitopes, and display different modes of action. Olokizumab that binds site III of IL-6 gives promising results in a phase IIb clinical trial for the treatment of RA [42], therefore it is particularly important to be able to determine patients for whom it would be effective.

There are still enough technical complications when working with circulating miRNAs as diagnostic and prognostic biomarkers. Among them are low miRNA abundance in plasma and serum, inaccurate miRNA quantification assays, lack of standardized methods for normalization, and there is a serious challenge to overcome them. However, despite of these limitations, investigation of epigenetic markers, including miRNAs, is undoubtedly a great achievement of molecular biology and important step to personalized medicine.

## 5. Conclusions

Our current work is the first study that links miRNA expression with the olokizumab treatment in Russian population of patients. Significantly differential expression of miR-26b, miR-29, miR-451, and miR-522 in the groups of responders and non-responders to olokizumab treatment, as well as their tight connection with IL-6/IL-6R signaling pathway make them a potential therapeutic response biomarker, which could be useful for personalized choice of therapy for RA patients.

## Figures and Tables

**Figure 1 jpm-10-00205-f001:**
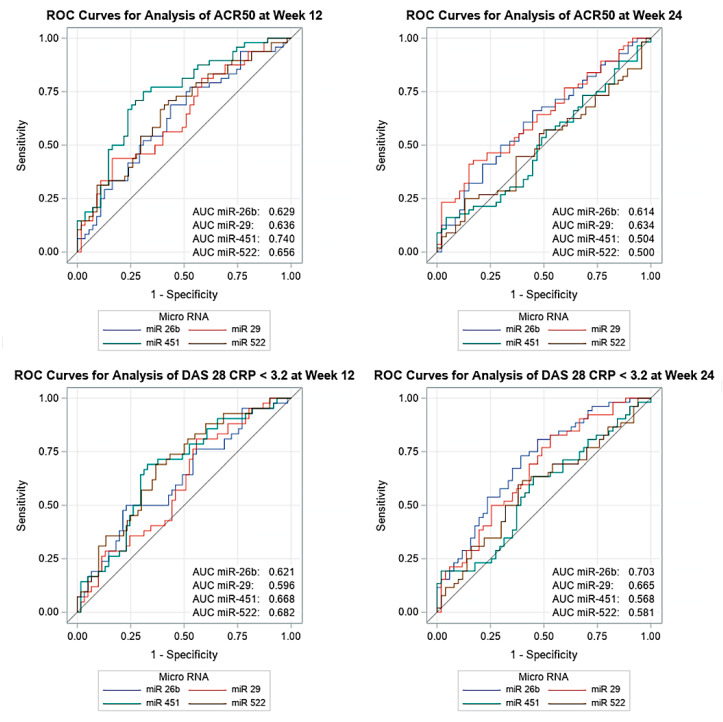
Associations between miRNAs miR-26b, miR-29, miR-451, miR-522 and ARC50, DAS28. The receiver operating characteristic (ROC) curves.

**Figure 2 jpm-10-00205-f002:**
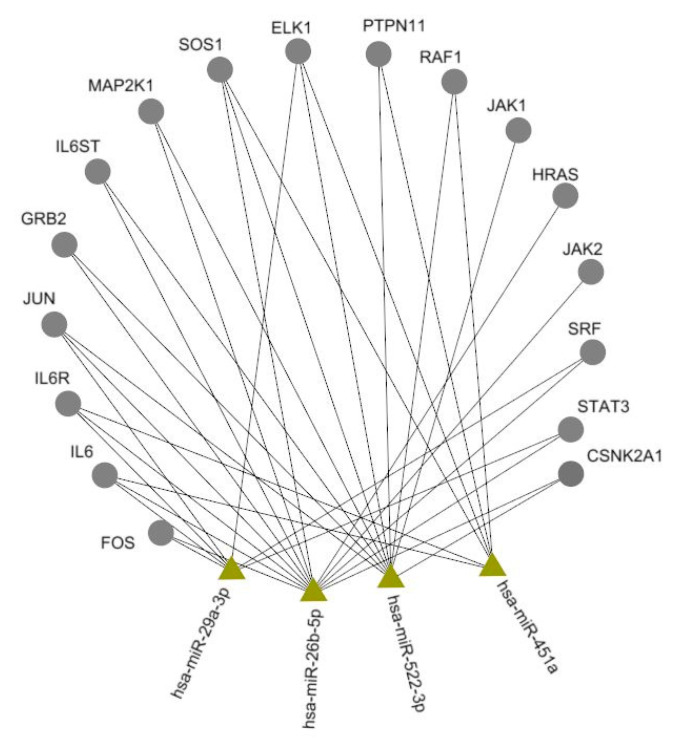
Associations of studied miRNAs with IL-6/IL-6R signaling pathway.

**Table 1 jpm-10-00205-t001:** Primer sequences.

miRNA	Sequence
miR-29	5′-GACTGATTTCTTTTGGTGTTCAAAAA-3′
miR-16	5′-TAGCAGCACGTAAATATTGGCA-3′
miR-155	5′-TTAATGCTAATCGTGATAGGGGTAAAA-3′
mir-20a	5′-TAAAGTGCTTATAGTGCAGGTAAAA-3′
miR-451	5′-AAACCGTTACCATTACTGAGTAAAA-3′
mir-26b	5′-GTTCAAGTAATTCAGGATAGGAAAA-3′
mir-522	5′-CTCTAGAGGGAAGCGCTTTCT-3′
miR-192	5′-CTGACCTATGAATTGACAGCAAA-3′
miR-137	5′-ACGGGTATTCTTGGGTGGATAA-3′

**Table 2 jpm-10-00205-t002:** Demographic and clinical characteristics of the patients.

Variables	Statistical Values	Patients (N = 103)
Age (years)	n mean ± SD min, max	103 51.8 ± 12.17 27, 82
Gender	n	103
Female Male	n (%) n (%)	90 (87.4%) 13 (12.6%)
Weight (kg)	n mean ± SD min, max	103 74.14 ± 15.515 41.0, 121.0
Disease duration (years)	n Median Q1, Q3	103 6.29 3.37, 11.20
Disease severity	n	101
Moderate (DAS28-CRP > 3.2 to ≤5.1) High (DAS28-CRP >5.1)	n (%) n (%)	7 (6.8%) 94 (91.3%)
DAS28-CRP	n mean ± SD min, max	101 6.03 ± 0.676 4.5, 8.1
CDAI	n mean ± SD min, max	101 40.47 ± 9.064 24.8, 69.3
HAQ-DI	n mean ± SD min, max	101 1.7339 ± 0.4882 0.500, 2.875
CRP (mg/mL)	n mean ± SD min, max	10,323.5 ± 22.85 1, 120
Anti-CCP (IU/mL)	n mean ± SD min, max	101 531.35 ± 671.132 0.4, 3408.4
RF (IU/mL)	n mean ± SD min, max	103 193.1 ± 224.15 7, 1250
Anti-CCP >10 IU/mL ≤10 IU/mL	n n (%) n (%)	101 86 (83.5%) 15 (14.6%)
RF ≥15 IU/mL <15 IU/mL	n n (%) n (%)	101 90 (87.4%) 13 (12.6%)
Basal Anti-CCP and RF Low/Low Medium High/High	n n (%) n (%) n (%)	101 15 (14.6%) 72 (69.9%) 14 (13.6%)

SD—standard deviation; CRP—C-reactive protein; Anti-CCP—Anti-cyclic citrullinated peptide antibodies; RF–Rheumatoid factor; DAS28–Disease Activity Score 28-joint Count; CDAI—Clinical Disease Activity Index; HAQ-DI–Health Assessment Questionnaire–Disability Index. N: number of patients in the study population. n: number of patients in the relevant group or the number of valid cases., Percentages are calculated as (100 × n/N). Q1 (Q3): 1st (3rd) quartile. Anti-cyclic citrullinated peptide antibodies: low level < 95.5 IU/mL, high level ≥ 669.3 IU/mL. Rheumatoid factor: low level < 44.0 IU/mL, high level ≥ 196.0 IU/mL. The threshold levels were determined by the tertiles of the corresponding distributions.

**Table 3 jpm-10-00205-t003:** Efficacy of olokizumab therapy.

Variables Weeks		Olokizumab (N = 103)
Response based on DAS28 (<3.2)		
Week 12 yes no Week 24 yes no	n (%) n (%) n (%) n (%)	103 42 (40.8%) 61 (59.2%) 103 52 (50.5%) 51 (49.5%)
Response based on ARC20		
Week 12 yes no Week 24 Yes no	n (%) n (%) n (%) n (%)	103 71 (68.9%) 32 (31.1%) 103 83 (80.6%) 20 (19.4%)
Response based on ARC50		
Week 12 yes no Week 24 yes no	n (%) n (%) n (%) n (%)	103 48 (46.6%) 55 (53.4%) 103 56 (54.4%) 47 (45.6%)

SD—standard deviation; CRP—C-reactive protein; ACR20 (ACR50)-American College of Rheumatology 20% (50%) improvement response criteria; DAS28—Disease Activity Score 28-joint Count. N: number of patients in the study population. n: number of patients in the relevant group or the number of valid cases. Percentages are calculated as (100 × n/N). Q1 (Q3): 1st (3rd) quartile.

**Table 4 jpm-10-00205-t004:** Relative expression of miRNAs at weeks 12 and 24 in comparison with baseline.

MiRNA	Olokizumab (N = 103)				
	Baseline Value *	Week 12 *	*p*-Value **	Week 24 *	*p*-Value **
miR-137	n = 101 −14.0901 −15.4189, −13.3029	n = 94 −13.8384 −14.7926, −12.9781	0.1741	n = 92 −14.1488 −15.3060, −12.8427	0.9689
miR-155	n = 103 −7.7179 −9.6218, −6.0038	n = 97 −6.6529 −8.7346, −5.3672	0.0123	n = 83 −6.8945 −8.9418, −5.6004	0.1211
miR-16	n = 103 −10.2718 −11.8904, −8.8278	n = 97 −9.3376 −10.9355, −8.2970	0.0001	n = 83 −9.6046 −11.3095, −8.3791	0.1147
miR-192	n = 102 −9.4246 −11.0052, −8.4285	n = 96 −8.8499 −10.5248, −8.0931	0.0151	n = 83 −9.0553 −10.8217, −7.8205	0.1326
miR-20a	n = 103 −12.5434 −13.5346, −11.2284	n = 97 −12.0229 −13.2547, −10.8907	0.0089	n = 83 −11.9538 −13.6044, −11.1414	0.9821
miR-26b	n = 103 −13.9983 −14.8531, −13.1991	n = 97 −14.0670 −14.7470, −13.1972	0.8457	n = 83 −14.3008 −15.2316, −13.4501	0.0291
miR-29	n = 103 −12.0607 −12.8295, −11.2436	n = 97 −11.8202 −12.7251, −11.1674	0.0204	n = 83 −12.0506 −12.8377, −11.2525	0.8998
miR-451	n = 103 −12.0970 −13.8915, −10.5405	n = 97 −11.3203 −12.9140, −9.2073	0.0018	n = 83 −11.1599 −13.1725, −9.5124	0.0234
miR-522	n = 102 −17.8232 −19.0515, −16.1847	n = 95 −17.9271 −18.9235, −16.6540	0.2977	n = 82 −18.3636 *** −19.3113, −17.2911	<0.0001

* The data presented as the median (Q1, Q3). Q1 (Q3): 1st (3rd) quartile. ** *p*-values are shown for comparing values at the corresponding time points with baseline using the Wilcoxon signed-rank test. *** The change comparing to week 12 was statistically significant (*p* = 0.0230). N: number of patients in the study population. The relative miRNA expression (dCt) was determined as Ct of the reference RNA (cel-miR-39-3p)—Ct of the studied miRNA. MiRNA values at week 24 were not included in the analysis for patients, who received additional DMARDs after week 14.

**Table 5 jpm-10-00205-t005:** Correlations between miR-26b, miR-29, miR-522 expression and disease severity and anti-CCP presence.

	Olokizumab (N = 103)	*p*-value
**Disease severity miR-26b**		0.0486
Moderate (DAS28-CRP >3.2 to ≤5.1)		
n	7	
Mean ± SD	−12.7001 ± 1.9300	
Median	−13.4802	
Q1, Q3	−13.9373, −12.2775	
Min, max	−14.2987, −8.6716	
High (DAS28-CRP > 5.1)		
n	94	
Mean ± SD	−14.0451 ± 1.2415	
Median	−14.0730	
Q1, Q3	−14.9105, −13.2551	
Min, max	−16.6515, −10.7522	
**Disease severity miR-29**		0.0141
Moderate (DAS28-CRP >3.2 to ≤5.1)		
n	7	
Mean ± SD	−10.8471 ± 1.5037	
Median	−11.2337	
Q1, Q3	−11.2997, −11.1783	
Min, max	−12.2331, −7.5453	
High (DAS28-CRP > 5.1)		
n	94	
Mean ± SD	−12.0729 ± 1.2746	
Median	−12.1499	
Q1, Q3	−12.8570, −11.4248	
Min, max	−14.9079, −8.2635	
**Anti-CCP miR-522**		0.0447
Positive		
n	85	
Mean ± SD	−17.3993 ± 2.1323	
Median	−17.6365	
Q1, Q3	−18.9118, −15.6329	
Min, max	−23.1875, −9.9698	
Negative		
n	15	
Mean ± SD	−18.5592 ± 1.3619	
Median	−18.1645	
Q1, Q3	−19.5899, −17.4640	
Min, max	−21.4611, −16.3436	

SD—standard deviation, Q1 (Q3): 1st (3rd) quartile. n: number of valid observations. *p*-values are shown for differences between clinical groups using the Mann-Whitney U test.

**Table 6 jpm-10-00205-t006:** MiRNAs included in the study and their role in RA.

miRNAs	Expression in RA	Localization	Target Genes	Potential Role in RA	Reference
miR-155	increased	peripheral blood mononuclear cell (PBMC)	*SOCS1*	inflammation and joint damage	[13]
	increased	synovial fibroblasts (SF)	*RIP1JKK*	NF-kB signaling pathway regulation	[20]
	increased	SF	*IKBKE*	decrease MMP3 expression, proliferation and FLS, inflammatory processes regulation	[14]
	increased	serum	*-*	inflammatory processes regulation	[21]
	increased	SF	*MMP-3, MMP-1*	blocks cytokine induction of MMP-1 and MMP-3	[22]
	increased	synovial macrophages, monocytes	*SHIP-1*	increase secretion of proinflammatory cytokines IL-6 and TNF-a	[23]
	decreased	whole blood	-	potential biomarker of methotrexate therapy response	[24]
miR-16	decreased	synovial fluid	-	NF-kB signaling pathway regulation	[25]
	decreased	serum	-	potential biomarker for RA diagnostic	[26,27]
	increased	SF	*Bcl-2*	apoptosis regulation	[17]
miR-20a	decreased	SF	*ASK1*	IL-6, TNF and IL-1b regulation, anti-inflammatory effect	[28]
	decreased	SF	*STAT3*	anti-inflammatory effect; suppress proliferation and induce apoptosis	[16]
	decreased	SF	*TXNIP*	regulate secretion of proinflammatory cytokines	[29]
miR-451	decreased	SF	*-*	regulate FLS proliferation and secretion of proinflammatory cytokines	[18]
miR-29	decreased	SF	*DNMT3a*	DNA methylation regulation	[30]
miR-192	decreased	SF	*CAV-1*	regulate proliferation and apoptosis	[19]
	decreased	SF	-	biomarker of therapy with TNF inhibitors response	[31]
miR-137	decreased	SF	*CXCL12*	regulate proliferation, migration and invasion	[32]
miR-26b	decreased	SF	*GSK-3P*	decrease proliferation and cytokine secretion, affecting signaling pathway Wnt/GSK-3b/b-katenin	[33]
miR-522	increased	SF	*SOCS3*	regulate expression of MMPs and proinflammatory cytokines	[15]

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
