# Peer review of "Analysis of miRNA Expression in Patients with Rheumatoid Arthritis during Olokizumab Treatment"

_jpm, 2020, doi:10.3390/jpm10040205_

Round 1

Reviewer 1 Report

This is an interesting article examining a battery of micro-RNAs involved in response of RA patients to treatment with an anti-IL6 antibody. The results demonstrate some significant alterations of certain micro-RNAs with presumed role in the IL-6 pathway although other biological processed are likely to be affected. The paper is well written but I have the following comments.

  1. Although the Authors state in the Introduction that the specific micro-RNAs were selected for their relevance to RA, they do not provide any data even briefly.
  2. In terms of the statistics, the Authors have conducted pairwise comparisons according to each time point. I would suggest a more robust approach involving GLM models with repeated measures followed by post hoc analysis for each time point.
  3. The results on the uni and muti-variable analysis should be put into a TABLE rather than being described in such great detail in the text.

Author Response

We thank the referee for the kind review and helpful comments.

Although the Authors state in the Introduction that the specific micro-RNAs were selected for their relevance to RA, they do not provide any data even briefly.

As it was recommended, the table with the role of investigated miRNAs in RA pathogenesis was included (revised parts of the text are marked in red).

In terms of the statistics, the Authors have conducted pairwise comparisons according to each time point. I would suggest a more robust approach involving GLM models with repeated measures followed by post hoc analysis for each time point.

Prior to the statistical analysis of data, statistical analysis plan was in place which defined the approaches to the statistical analysis. In general, microRNA data were planned to be analyzed using non-parametric methods. Taking that into consideration, GLM model was not implemented (though we recognize the robustness of this method when applied to data with moderate deviations from the assumption of normality). The non-parametric counterpart of GLM model would be Friedman test, which was considered but not implemented either, since it was known that some of the subjects had measurements only for a subset of time points. Running a Friedman test on such data set would result in exclusion of all subjects with incomplete data from the analysis.

Though we recognize the importance of controlling the type I error rate in presence of multiple comparisons it was decided that it was also important to maintain as much valuable information as possible in the statistical analysis considering its exploratory nature.

The results on the uni and muti-variable analysis should be put into a TABLE rather than being described in such great detail in the text.

The table with the results of the uni and muti-variable analysis are included into the Supplementary (Supplementary 2).

Reviewer 2 Report

  • line 85: patients were considered to have failed MTX after 12 weeks of treatment, was it only subjective or was there objective evidence? better to have CDAI or DAS28 or at least lack of ESR/CRP improvement.
  • - Patients who received olokizumab were also on MTX, how can we determine that those effects were not as a result of MTX, would favor having a control study.

Author Response

We thank the referee for the comments and questions. It was very important and helpful indications.

line 85: patients were considered to have failed MTX after 12 weeks of treatment, was it only subjective or was there objective evidence? better to have CDAI or DAS28 or at least lack of ESR/CRP improvement.

Inadequate response to MTX therapy was defined as a subject with at least 12 weeks of exposure prior to Screening and with either: a) Absence of any documented clinically significant response; or b) Documented initial clinical response with subsequent loss of that response or partial response. A patient eligible for inclusion in the study had to have moderately to severely active RA disease as defined by all of the following: a. ≥6 tender joints (68-joint count) at Screening and baseline; and b. ≥6 swollen joints (66-joint count) at Screening and baseline; and c. CRP above ULN at Screening based on the central laboratory results. As a result, all subjects of the core study had DAS28(CRP)>3.2 and more than 80% had DAS28 (CRP) >5.1 at baseline, minimum SDAI and CDAI values at baseline were 17.1 and 16.6 respectively.

Patients who received olokizumab were also on MTX, how can we determine that those effects were not as a result of MTX, would favor having a control study.

We thank the referee for drawing our attention to that. There was a mistake in the text and actually patients involved in the CREDO 1 (CL04041022) project didn’t get MTX during the olokizumab treatment. We corrected it in the manuscript. Therefore, a control study is not relevant.

Round 2

Reviewer 1 Report

I am content with the revised manuscript.

Author Response

Thank you. 

Reviewer 2 Report

since patients weren't on methotrexate, why were they receiving folic acid weekly while on Olokizumab? Thanks!

Author Response

We are very grateful to the referee for the comment. The full sentence with both methotrexate and folic acid had to be deleted already after the first correction, but occasionally a part of it remained. We apologize for this mistake. Please find attached a correct version of the manuscript.